# Characterization of Saponins from Various Parts of *Platycodon grandiflorum* Using UPLC-QToF/MS

**DOI:** 10.3390/molecules27010107

**Published:** 2021-12-24

**Authors:** So-Jeong Lee, Heon-Woong Kim, Suji Lee, Ryeong Ha Kwon, Hyemin Na, Ju Hyung Kim, Chi-Do Wee, Seon Mi Yoo, Sang Hoon Lee

**Affiliations:** Department of Agro-food Resources, National Institute of Agricultural Science, Rural Development Administration, Wanju-gun 55365, Jeollabuk-do, Korea; quizhalo@naver.com (S.-J.L.); ksharrier@korea.kr (H.-W.K.); sujiseven@naver.com (S.L.); haha9733@korea.kr (R.H.K.); na970713@naver.com (H.N.); ddong128@naver.com (J.H.K.); cdwee@korea.kr (C.-D.W.); yousm@korea.kr (S.M.Y.)

**Keywords:** *Platycodon grandiflorum*, saponin, UPLC-QTOF/MS

## Abstract

*Platycodon grandiflorum* (PG) is known as a high-potential material in terms of its biological activity. The objective of this report is to provide chromatographic and mass fragment ion data of 38 simultaneously identified saponins, including novel compounds, by analyzing them through ultra-performance liquid chromatography coupled with quadrupole time-of-flight mass spectrometry (UPLC-QToF/MS). In so doing, we investigated their diverse conditions, including morphological parts (stems, roots, buds, and leaves), peeling (or not), and blanching of PG. The total contents of individual saponins indicated an order of roots (containing peel, 1674.60 mg/100 g, dry weight) > buds (1364.05) > roots (without peel, 1058.83) ≈ blanched roots (without peel, 945.17) ≈ stems (993.71) ≈ leaves (881.16). When considering three types of aglycone, the platycodigenin group (55.04 ~ 68.34%) accounted for the largest proportion of the total content, whereas the platycogenic acid A group accounted for 17.83 ~ 22.61%, and the polygalacic acid group represented 12.06 ~ 22.35%. As they are classified as major compounds, novel saponins might be utilized for their role in healthy food for human consumption. Additionally, during blanching, the core temperature of PG was satisfied with the optimal condition, thus activating the enzymes related to biotransformation. Furthermore, through the use of this comprehensive data, additional studies related to buds, as well as roots or the characterization of individual saponins, can be conducted in a rapid and achievable manner.

## 1. Introduction

*Platycodon grandiflorum* belongs to the Campanulaceae family, and it has been widely used as a traditional herb medicine to treat cough and sore throat; it has also been used as a food source, especially in northeast Asian countries, such as South Korea and China [1,2,3,4,5,6]. Raw PG roots are consumed in salads, whereas pickled fresh roots are used in juice, the traditional dish kimchi, jeonggwa, and as a functional food taken in pill form, or cooked by blanching or steaming [1,7,8]. *Platycodi Radix* (PR), i.e., the roots of PG, have reported biological properties, such as anti-inflammatory, antitumor, hepa protection, and antiobesity effects [9,10,11,12,13,14].

Plants synthesize different bioactive secondary metabolites such as saponins, alkaloids, flavonoids, terpenoids, and tannins. Among these, saponins, which are derived from ‘*sapo*’, a Latin word meaning ‘soap’, are largely subdivided into two classes: a triterpenoid (30 carbon atoms) and a steroid (27 carbon atoms). Additionally, based on the carbon of the aglycone, saponins are classified into 12 types, including dammaranes and oleananes [15,16,17,18]. Saponin derivatives are found in a large number of plants, such as *Glycine max*, *Panax ginseng*, *Codonopsis lanceolata*, *Camellia sinensis*, *Puerariae lobata*, and *Centella asiatica*, including *Platycodon grandiflorum* [19,20,21,22,23]. Previous reports show the total saponin content of *Panax ginseng* to be 57.2 ~ 121.8 mg/g, dry weight (DW) [24,25]. Until recently, saponins of *Platycodon grandiflorum* have been reported to have biological activities related to anti-inflammation in certain respiratory diseases. These corresponding symptoms become influential by spreading coronavirus (COVID-19) around the world, a disease that can have a lethal effect on the respiratory system. Therefore, this study focuses on analyzing saponins in *Platycodon grandiflorum* to support follow-up studies.

Many previous studies reported the characterization of saponins from *Platycodi Radix* using HPLC (high performance liquid chromatography) [9,26,27,28,29]. In recent years, UPLC (ultra-performance liquid chromatography) coupled with MS (mass spectrometry) has shown itself to be an appropriate instrument to analyze various saponin derivatives [12,30,31,32,33]. Based on a previous study, the identification and quantification of analytes using an MRM (multiple reaction monitoring) mode has been implemented to achieve a higher sensitivity [34].

In terms of previous research, there has yet to be a report comprehensively describing the four different parts of the plant, including the stems, leaves, roots, and buds, as well as a description of the materials used for blanching. Even though other studies have been conducted with various parts of PG, no study has simultaneously analyzed 38 saponins using UPLC-QTOF-mass spectrometry [4,14,32,35,36]. Therefore, in this study, using UPLC combined with QTOF-mass spectrometry analysis, 38 saponins were identified or tentatively characterized with four different parts, including the blanching process. Thus, considering the change in saponin content in these materials, along with the transformation in their structure, platycodin (major saponin in PG) pathways are examined in light of previous studies [37,38,39,40].

## 2. Results and Discussion

### 2.1. Identification of Saponins from P. grandiflorum

In this study, the identification of saponin derivatives was compared with the nine correct standards by retention time (RT), whereas mass fragmentation was measured using the MRM (multiple reaction method). The standards were deapi-platycoside E, platycoside E, deapi-platycodin D3, platycodin D3, deapi-platycodin D (not detected in samples), platycodin D2, platycodin D, polygalacin D (not detected), and 2”-*O*-acetyl platyconic acid A, respectively, in the elution order of RT. As a result, peak 2, 3, 8, 10, 24, 25, and 38 were confirmed accurately and identified as deapi-platycoside E, platycoside E, deapi-platycodin D3, platycodin D3, platycodin D2, platycodin D, and 2”-*O*-acetyl platyconic acid A, respectively. The remains of the peaks were identified by comparing the mass fragmentation and RT of the compounds in a previously constructed saponin library (Appendix A). A total of 38 saponin derivatives, including major saponins (platycodin D and platycodin E known as PG saponins), were identified from *P. grandiflorum* using ultra-performance liquid chromatography coupled with electrospray ionization tandem mass spectrometry (UPLC-ESI-MS) (Appendix A).

### 2.2. Quantification of Saponins from P. grandiflorum

The contents of platycodin-related saponins in *Platycodon grandiflorum* were calculated using tubeimoside I (100 ppm) as an internal standard. The 38 saponins we identified were divided into three types in the aglycone group (Figure 1). Total saponin content is presented in Table 1, and the contents of the four parts ranged from 881.16 ± 5.15 to 1674.60 ± 25.45 mg/100 g dry weight (DW) in *Platycodon grandiflroum*. The chromatograms of representative samples are presented in Figure 2 and Figure 3.

The results of total saponin contents of the individual parts were in the following order: roots (containing peel, 1674.60 mg/100 g, dry weight (DW)) > buds (1364.05) ≈ roots (without peel, 1058.83) ≈ stems part (993.71) ≈ blanched roots (without peel, 945.17) ≈ leaves (881.16). This result indicated that the total saponin levels were diverse according to individual parts of PG (*p* < 0.05), except for between the stem and root parts (without peel) (*p* > 0.05). As the ability to remove the PG peel was hindered, the total saponin content was significantly reduced (*p* < 0.05). This result was assumed to be due to the fact that the peel comprised a considerable number of saponins. In addition, the aerial parts, including stems and leaves, contained phenolic acid or flavonoids, instead of saponins, which induced a difference in the total saponin content between the roots and other parts [35,41]. Previous studies reported that the total contents of individual saponin derivatives were 6410 ~ 13,197 (mg/100 g, DW) by quantifying based on corresponding external standards using HPLC [39,42,43]. Despite the use of external standards, an accurate confirmation of those compounds may be necessary for selecting authentic standards. HPLC could be limited in distinguishing simultaneously eluted saponins based on retention time. Other reports suggested that the total contents of crude saponins were 2524 and 23,400 (mg/100 g, DW) [44,45]; the analysis method for crude saponins induces a large difference in contents. In addition, quantification based on the internal standard, such as the method used in this study without external standards, could generate inaccurate data in terms of determining saponin contents only. Consequently, this study focused on the accurate identification of individual saponins using UPLC-QToF/MS, which is known as an instrument that indicates high resolutions, and then illustrated the proportion of individual saponin derivatives by comparing each sample in reference to quantification data based on the internal standard. In the case of stem parts, the deapi-platycodin D3 (peak 8, 0.13%) indicated the lowest proportion (%) of the total content, whereas 3”-*O*-acetyl platycurodin D (peak 23, 8.32%) was the highest proportion among the constituents. The two saponins are based on the aglycone of platycodigenin (*m/z* 520). Accordingly, numerous studies using LC–MS show that deapi-platycodin D3 is clearly a minor compound [37,46,47]. The observed result of the bud parts was 2”-*O*-acetyl polygalacin D3 (peak 21, 0.36%), which presented the lowest proportion (%); this structure also belongs to the polygalacic acid (*m/z* 504) group. In contrast, as based on platycogenic acid A (*m/z* 534), the novel 2”-*O*-acetyl platyconic acid A (peak 38, 10.25%) showed the highest proportion (%) of the total content. In the leaves, 2”-*O*-acetyl platyconic acid A (peak 38, 11.55%) was revealed as having the most obvious intensity, with the same in the bud case; conversely, the deapi-platycodin D2 (peak 19, 0.37%) showed the lowest concentration of intensity. Additionally, these compounds are associated with the aglycone of platycogenic acid A (*m/z* 534) and platycodigenin (*m/z* 520), respectively. Overall, in stems, buds, and leaves, the novel 2”-*O*-acetyl platyconic acid A (peak 38, 10.28, 10.25, 11.55%) showed the most major saponin concentrations; this compound is based on the aglycone structure of platycogenic acid A (*m/z* 534). The result of the roots (containing peel) was tentatively assigned as 3”-*O*-acetyl platyconic acid A3 (peak 13, 0.11%), which indicated the lowest intensity and was based on platycodigenin (*m/z* 520) aglycone; however, 3”-*O*-acetyl platycodin D (peak 27, 8.62%) showed the highest concentration. After removing the peel of PG, the highest and lowest compounds were platycoside E (peak 3, 9.62%) and 3”-*O*-acetyl platycoside E (peak 5, 0.02%), and both were related to platycodigenin (*m/z* 520) aglycone in this sample. Ultimately, saponins showing predominant contents in each sample were in the aglycone of the group in platycodigenin (*m/z* 520), followed by platycogenic acid A (*m/z* 534).

In addition, among the aglycone groups, the triterpenoid saponin related to platycodigenin (55.04–68.34%) was significantly more abundant than platycogenic acid A (17.83–22.61%) and polygalacic acid (12.06–22.35%). The compounds were related to the aglycone of platycodigenin (*m/z* 520), and 10 saponins of the 22 derivatives belonged to this group, indicating their significant difference (*p* < 0.05). They are the following compounds: platycoside G2 (peak 1), platycoside G1 (peak 2), platycoside E (peak 3), 3”-*O*-acetyl platycoside E (peak 5), platycoside P (peak 6), deapi-platycodin D3 (peak 8), 2”-*O*-acetyl platycoside E (peak 9), deapi-2”-*O*-acetyl platycodin D3 (peak 15), deapi-platycodin D2 (peak 19), and 2”-*O*-acetyl platycurodin D (peak 33); these do not include the undetected saponins. Additionally, in the platycogenic acid A (*m/z* 534) group, 2”-*O*-acetyl platyconic acid C (peak 29) and platyconic acid A (peak 31) indicated significant differences (*p* < 0.05). The remaining 3”-*O*-acetyl platyconic acid A (peak 32) and 2”-*O*-acetyl platyconic acid A (peak 38) groups showed no meaningful differences (*p* > 0.05). Moreover, in the case of polygalacic acid (*m/z* 504), only four compounds, including platycoside D (peak 4), deapi-polygalacin E (peak 7), 3”-*O*-acetyl polygalacin D3 (peak 16), and 2”-*O*-acetyl polygalacin D3 (peak 21), indicated meaningful differences (*p* < 0.05), whereas the others in this group were compounds that showed no significant differences (*p* > 0.05).

By focusing on these novel compounds in the aglycone group of polygalacic acids, deapi-polygalacin E (peak 7, 3.01%) was detected as a significant proportion of the total saponin contents in the stems, which was comparable to other parts. Additionally, 2”-*O*-acetyl platyconic acid A3 (peak 18, 4.54%) was predominantly indicated in the buds. In the case of the platycogenic acid A group, 2”-*O*-acetyl platyconic acid A (peak 38, 8.02 ~ 11.55%) and 3”-*O*-acetyl platyconic acid A (peak 32, 7.58 ~ 11.06%) were detected as major saponins. Finally, the results of platycodigenin aglycone group showed that platycurodin D (peak 20) occupied a significant proportion of the stems and buds, representing 6.54% and 5.03% of the total contents, respectively.

Previous studies reported that platycodin D or polygalacin D were the most dominant saponin derivative, and the contents of each compound indicated values as diverse as 246.2 ~ 1196.9 (mg/100 g, DW) and 191.6 ~ 1241.0 (mg/100 g, DW) when changing soil conditions, dividing the root, and changing the cultivated region [31,48,49,50,51,52,53]. However, Lu et al. showed that for the individual saponin content of various cultivated regions, some provinces revealed platycoside E (69.1 ~ 346.2 mg/100 g, DW) to be the most prevalent compound; however, other regions showed platycodin D (40.9 ~ 511.3 mg/100 g, DW) to be the predominant compound [42]. Therefore, these results indicated that the content and composition of individual saponins could differ dramatically by region. Many previous reports suggested that platycodin D and polygalacin D were major compounds. However, this study showed that 2”-*O*-acetyl platyconic acid A (peak 38, 101.78 ~ 139.78 mg/100 g, DW), a tentatively identified novel saponin, was detected as the most prevalent compound in the stem, bud, and leaves of PG using UPLC-QToF/MS. Additionally, the predominant saponins in the roots containing peel were 3”-*O*-acetyl platycodin D (peak 27, 144.40 mg/100 g, DW) and platycoside E (peak 3, 101.90 mg/ 100 g, DW) in the roots without the peel, respectively.

With reference to the table detailing saponin contents in roots (without peel) and blanched roots (without peel), the total saponin was reduced from 1058.83 (mg/100 g, DW) to 945.17 (mg/100 g, DW) after blanching (Table 1). Additionally, in the polygalacic acid group (*m/z* 504), one of the three groups, tentatively identified as novel compound 3”-*O*-acetyl platyconic acid A3 (peak 13), was removed entirely due to blanching. The proportion of polygalacin D3 (peak 14), 2”-*O*-acetyl polygalacin D (peak 37), and 3”-*O*-acetyl polygalacin D2 (peak 28) on the total saponin in each sample decreased after the process. Conversely, the contents of 3”-*O*-acetyl polygalacin D3 (peak 16), 2”-*O*-acetyl polygalacin D3 (peak 21), 3”-*O*-acetyl polygalacin D (peak 30), and 2”-*O*-acetyl polygalacin D2 (peak 36) were higher than before blanching. These results indicate that acetylation from polygalacin D3 to two compounds, including 3”-*O*-acetyl polygalacin D3 (peak 16) and 2”-*O*-acetyl polygalacin D3 (peak 21), might occur during the blanching process. Additionally, the trans-acetylation from 2”-*O*-acetyl polygalacin D (peak 37) to 3”-*O*-acetyl polygalacin D (peak 30), as well as from 3”-*O*-acetyl polygalacin D2 (peak 28) to 2”-*O*-acetyl polygalacin D2 (peak 36), was assumed to occur simultaneously due to the migration between 3”-*O*-acetyl and 2”-*O*-acetyl by each specific enzyme. In the case of the platycogenic acid A group (*m/z* 534), platyconic acid A (peak 31) decreased after the process, whereas the proportional contents of 3”-*O*-acetyl platyconic acid A (peak 32) and 2”-*O*-acetyl platyconic acid A (peak 38) increased. Finally, in the platycodigenin group (*m/z* 520), platycoside G2 (peak 1) lacked xyloside from platycoside G1 (peak 2), and was reduced in the blanching procedure; in contrast, the latter increased. By the mechanism of de-xylosylation and acetylation, deapi-platycodin D2 (peak 19) might be converted to corresponding platycoside P (peak 6) and deapi-3”-*O*-acetyl platycodin D2 (peak 22), respectively. Similarly, the phenomenon of decreasing platycurodin D (peak 20), platycodin D (peak 25), and platycodin D2 (peak 24) was presumed to change into 3 or 2”-*O*-acetyl platycurodin D (peak 23, 33), 3 or 2”-*O*-acetyl platycodin D (peak 27, 35), and 2”-*O*-acetyl platycodin D2 (peak 34) during the process with the use of acetic acid. Additionally, 2”-*O*-acetyl platycodin D2 (peak 34) might be converted from 3”-*O*-acetyl platycodin D2 (peak 26) by the migration of the acetic acid position from 3-OH to 2-OH by a trans-acetylation mechanism. Previous studies reported that specific enzymes have the ability to transform from platycoside E (peak 3) and platycodin D3 (peak 10) to platycodin D (peak 25) through the removal of glucoside [2,53,54]. Likewise, xylosylation was predicted to occur with a similar mechanism during the blanching process. Additionally, in the case of astragaloside from *Radix Astragali* in the saponin group, the optimal temperature for acetyl estrase’s activity was 35 °C, and it was stable at temperatures lower than 45 °C [55]. Nyakudya et al. noted the platycodin pathway in the enzymatic transformation by human intestinal microorganisms; this finding reinforces the acetylation mechanism, as well as de-glycosylation, including glucose and xylose, in PG [39]. Ma et al. reported that the cytochrome P450 (CYPs), polygalacic acid, platycodigenin, and platycogenic acid A are converted by one another; they also found that glycosyl transferase affects the detachment of carbohydrates in PG [38]. Therefore, in this study, using a blanching procedure, the mechanism of acetylation might have occurred in the provided temperature (35 ~ 45 °C) over a short time (2 min) because the core temperature of PG is satisfied with these specific conditions. Additionally, de-xylosylation could be illustrated in the phenomenon of de-glycosylation.

In all of the published literature, it is clear that *Platycodon grandiflorum* saponins (PGS) are oleanane types of triterpenoids [46,56]. PGS have hepato-protective effects by blocking the bioactivation of CCl_4_ and scavenging free radicals while inhibiting CYP2E1 activity [57]. In addition, PGS can also be immunological adjuvants. One of the PGS, platycodin D2, has potential to be a safe adjuvant that increases Th1 and Th2 cytokines [35,58]. Platycodin D could be used for the treatment of cancer via the induction of apoptosis in human leukocytes [11]. Moreover, it may be a potent adjuvant for increasing immune responses against the hepatitis B antigen [59]. 2”-*O*-acetyl polygalacin D2 and platycodin D were reported to reduce neuroinflammation [60]. Platycodin D3 regulates the production of NO (nitric oxide) and TNF-α, resulting in inflammatory pulmonary diseases [61,62]. Furthermore, a previous study reported a relationship between structure and function on seven major saponins from PG, and the result of this study was that platycodin D could be a major constituent for its strong immunological adjuvant properties and hemolytic activity [63].

In the above mentioned reports, major saponins, such as platycodin D and polygalacin D, were detected by a comparatively less accurate method: liquid chromatography (LC). This method is less accurate than LC coupled mass spectrometry (MS), which was used in this study. The results of this study provide the chromatographic and mass fragment data of individual saponins, tentatively identified by an accurate method: UPLC-QToF/MS. Accordingly, the compounds detected by this method include 2”-*O*-acetyl platyconic A (peak 38), 3”-*O*-acetyl platycodin D (peak 27), and platycoside E (peak 3) as the most prevalent constituents, which themselves could be separated and purified to produce health functional foods; moreover, the significantly large proportion of 2”-*O*-acetyl platyconic A could be used in various industries. In addition, in terms of their meaningful difference in saponin content, buds may be utilized in health functional foods.

## 3. Materials and Methods

### 3.1. Chemicals and Reagents

Methanol and water (HPLC grade) were purchased from Fisher Scientific (Fair Lawn, NJ, USA). Deapi-platycoside E, platycodin D3, platycodin D, and polygalacin D were purchased from Target Molecule Corporation (Boston, MA, USA), and platycoside E, platycodin D2, deapi-platycodin D3, and deapi-platycodin D were purchased from Med Chem Express (Monmouth, NJ, USA), whereas 3”-*O*-acetylplatyconic acid A from ALB Tech (Albany, NY, USA) was used as an external standard for identification. Tubeimoside I was used as the internal standard (100 ppm).

### 3.2. Plant Materials

Roots of *P. grandiflorum* were collected from Hoengseong, Korea in 2020. Leaves, stems, and buds of *P. grandiflorum* were obtained from Icheon, Korea in 2019. One part of the roots was blanched at 100 °C for 2 min after being washed with water, and the remains were only washed without blanching. Then, all samples were freeze-dried, pulverized, and stored in a deep freezer before analysis.

### 3.3. Extraction of Saponins

Then, 0.1 g of powered samples was extracted, first with 1.5 mL, and then with 1 mL extraction solvent (70% methanol). The mixture of the two extractions was vortexed, centrifuged for 10 min at 13,000 rpm, 10 °C, and the supernatant was filtered using a syringe filter (PVDF 0.2 µm, 13 mm, Whatman, Kent, England). The saponin extract was concentrated using N_2_ gas and redissolved with 15 mL of water. A solid-phase extraction method using a Hypersep C18 cartridge (Thermo Scientific, Rockwood, TN, USA) was initiated to isolate saponin derivates from the extract. The cartridge was activated with methanol (3 mL), followed by water (6 mL) for conditioning. The redissolved saponin extract was loaded on the cartridge, and then the diluted solution containing ISTD (internal standard; 100 ppm) was loaded on the cartridge. After washing the cartridge with water (6 mL), the crude saponin extract was eluted by methanol (15 mL) and concentrated using N_2_ gas. The final extract was redissolved with 500 µL of the extraction solvent (70% methanol) and filtered by a syringe filter (PVDF 0.2 µm, 13 mm, Whatman, Kent, England).

### 3.4. UPLC-DAD-QToF/MS Analysis

Saponin derivative profiling was performed by UPLC-DAD-QToF/MS (Sciex Co., Framingham, MA, USA) (Table 2 and Table 3). Identification and quantification of saponins were conducted by an analysis software version (Sciex Co., USA).

### 3.5. Identification and Quantification of Saponin Derivatives

The saponin library for *Platycodon grandiflorum* was constructed by data from the previous literature, as confirmed by MS and NMR analyses (Appendix A). Saponin derivatives were identified by the data of these libraries, comprising compound names, used parts, molecular weight, and MS fragment ion patterns. Based on 9 standards confirmed and selected by multiple reaction monitoring (MRM) of *Platycodon grandiflorum*, 39 saponin derivatives were identified. Quantification was implemented by calculating the relative peak area of compounds compared with the internal standard (tubeimoside I, 100 ppm).

### 3.6. Statistical Analysis

One-way ANOVA was performed to confirm a significant difference between individual averages using Duncan’s multiple range test (*p* < 0.05) in SPSS (version 25.0, SPSS Inc., Chicago, IL, USA).

## 4. Conclusions

The objective of this study was to provide comprehensive data related to the chromatographic and mass fragment ion patterns of the 38 saponin derivatives in relation to four parts of *Platycodon grandiflorum*: stems, roots, leaves, and buds. Our method was based on a library constructed of the findings from previous literature, and newly detected data in this study allowed us to tentatively identify novel compounds to support further studies. In this study, except for the well-known major saponins, such as platycodin D, tentatively identified novel saponins indicated larger peaks than those in the abovementioned major compounds. Though the roots of PG have been comprehensively studied, studies on the buds and individual saponins, including novel and minor compounds, remain necessary; furthermore, their activities in relation to human health should also be studied. In conclusion, we provided chromatographic and mass fragment ion pattern data of 38 saponins from four parts of PG; peeling (or not) and blanching may be used in diverse experiments related to their characterization, and may aid in the production of functional food for human consumption.

## Figures and Tables

**Figure 1 molecules-27-00107-f001:**
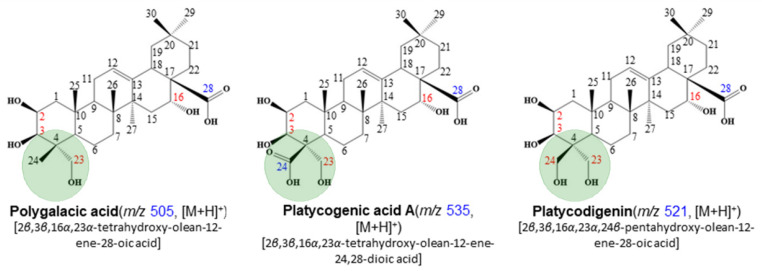
Three types of aglycone groups (polygalacic acid, platycogenic acid A, platycodigenin).

**Figure 2 molecules-27-00107-f002:**
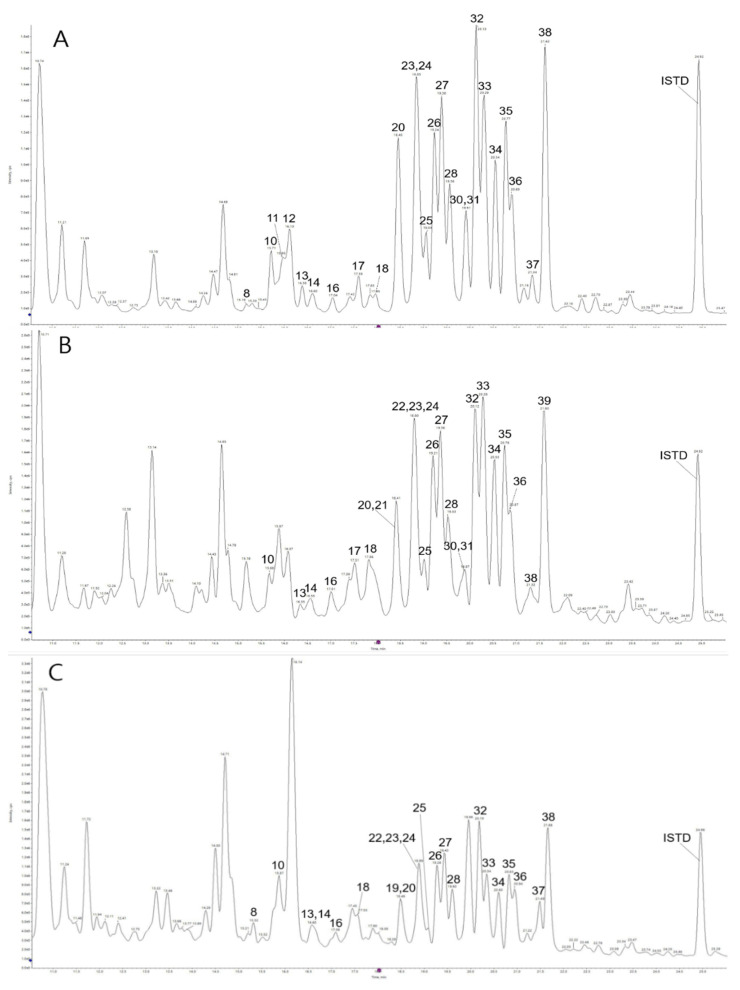
UPLC chromatograms of saponins from *Platycodon grandiflorum.* (**A**) stems part, (**B**) buds parts, and (**C**) leaves part.

**Figure 3 molecules-27-00107-f003:**
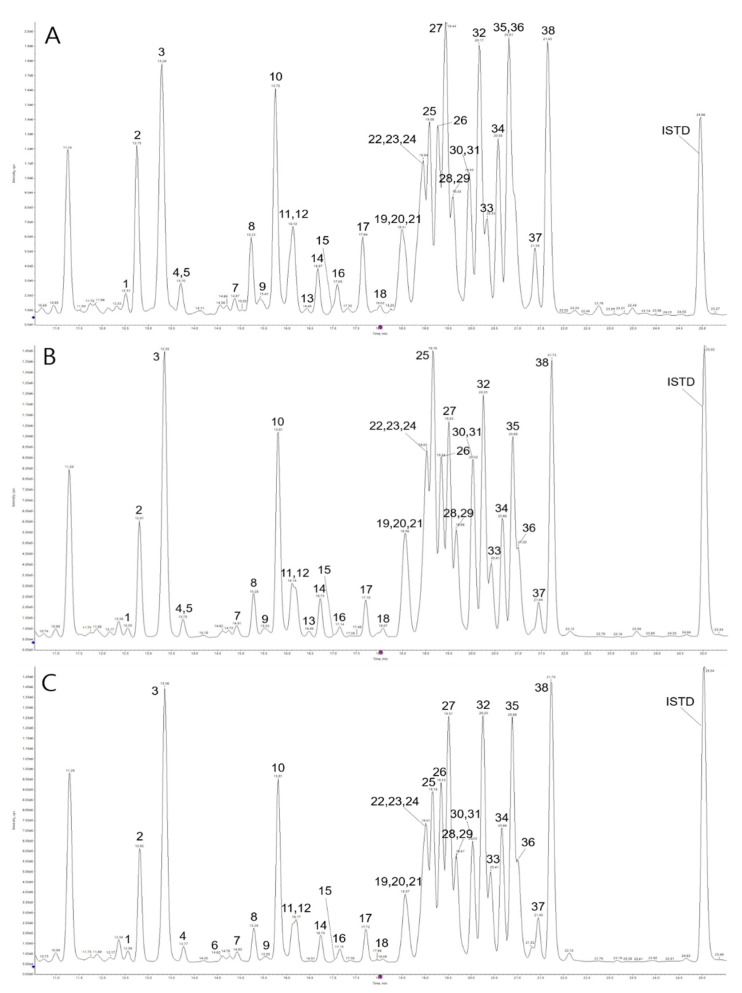
UPLC chromatograms of 38 saponins from *Platycodon grandiflorum.* (**A**) roots (containing peel), (**B**) roots (without peel), and (**C**) blanched roots (without peel).

**Table 1 molecules-27-00107-t001:** Saponin derivatives contents (mg/100 g, DW) of *P. grandiflorum.* Each value was calculated as mean ± SD (n = 3) using the internal standard (tubeimoside I 100 ppm); ND means “not detected”; contents were expressed in mg/100 g DW (dry weight). Different small letters with mean values (n = 3) reveal a significant difference (at *p* < 0.05) by Duncan’s multiple range test.

Aglycones	Peak No. ^(1)^	Stems	Buds	Leaves	Roots (Containing Peel)	Roots (without Peel)	Blanched Roots (without Peel)
**Polygalacic acid** **(*m/z* 504)**	4	ND	ND	ND	11.97 ± 0.35 ^a^	5.88 ± 0.49 ^b^	4.60 ± 0.34 ^c^
	7	ND	ND	ND	7.14 ± 0.15 ^a^	3.95 ± 0.19 ^b^	3.15 ± 0.06 ^c^
	11	29.89 ± 3.67 ^a^	ND	ND	12.19 ± 0.70 ^b^	11.56 ± 1.06 ^b^	7.47 ± 0.45 ^c^
	13	10.54 ± 1.30 ^a^	6.53 ± 0.56 ^b^	4.85 ± 2.55 ^b^	1.90 ± 0.20 ^c^	2.02 ± 0.49 ^c^	ND
	14	11.51 ± 2.02 ^b^	18.34 ± 3.25 ^a^	18.11 ± 1.94 ^a^	18.80 ± 0.29 ^a^	11.68 ± 0.56 ^b^	7.51 ± 0.16 ^c^
	16	6.41 ± 0.97 ^d^	15.46 ± 1.48 ^b^	17.56 ± 1.94 ^a^	8.98 ± 0.30 ^c^	2.52 ± 0.40 ^e^	2.93 ± 0.23 ^e^
	18	9.02 ± 1.82 ^bc^	61.97 ± 11.76 ^a^	16.37 ± 4.79 ^b^	3.83 ± 0.48 ^c^	3.17 ± 0.41 ^c^	0.97 ± 0.16 ^c^
	21	ND	4.96 ± 0.28 ^b^	ND	6.18 ± 0.30 ^a^	2.00 ± 0.24 ^d^	2.43 ± 0.06 ^c^
	28	44.36 ± 2.86 ^b^	65.53 ± 7.09 ^a^	47.98 ± 3.28 ^b^	42.28 ± 4.49 ^b^	29.96 ± 1.31 ^c^	26.22 ± 0.51 ^c^
	30	13.22 ± 1.40 ^c^	16.94 ± 3.76 ^b^	ND	35.96 ± 0.75 ^a^	12.39 ± 1.30 ^c^	16.37 ± 0.45 ^b^
	36	37.51 ± 1.04 ^c^	57.12 ± 7.97 ^a^	46.63 ± 5.02 ^b^	55.71 ± 1.33 ^a^	29.99 ± 6.74 ^c^	32.61 ± 0.52 ^c^
	37	14.01 ± 2.33 ^cd^	19.72 ± 7.81 ^c^	45.45 ± 3.09 ^a^	26.73 ± 2.40 ^b^	12.61 ± 1.48 ^d^	12.09 ± 0.41 ^d^
**Subtotal**		**176.48 ± 11.66 ^c^**	**266.56 ± 26.94 ^a^**	**196.95 ± 11.93 ^c^**	**233.41 ± 4.97 ^b^**	**127.73 ± 12.26 ^d^**	**116.33 ± 1.39 ^d^**
**Platycogenic acid A** **(*m/z* 534)**	29	ND	ND	ND	9.09 ± 2.05 ^a^	6.95 ± 0.41 ^b^	5.95 ± 0.23 ^b^
	31	17.15 ± 2.60 ^c^	11.55 ± 2.84 ^d^	ND	28.25 ± 1.84 ^b^	43.43 ± 3.53 ^a^	18.90 ± 1.07 ^c^
	32	100.99 ± 7.24 ^b^	125.92 ± 4.94 ^a^	97.42 ± 3.12 ^b^	126.91 ± 3.01 ^a^	80.98 ± 7.62 ^c^	85.44 ± 5.35 ^c^
	38	102.18 ± 1.51 ^b^	139.78 ± 12.81 ^a^	101.78 ± 5.71 ^b^	134.30 ± 0.91 ^a^	87.79 ± 3.14 ^c^	90.84 ± 1.31 ^c^
**Subtotal**		**220.32 ± 9.73 ^c^**	**277.24 ± 15.34 ^b^**	**199.20 ± 8.77 ^d^**	**298.55 ± 2.83 ^a^**	**219.15 ± 8.82 ^c^**	**201.13 ± 5.47 ^d^**
**Platycodigenin (*m/z* 520)**	1	ND	ND	ND	8.69 ± 0.05 ^a^	2.82 ± 0.12 ^c^	3.07±0.19 ^b^
	2	ND	ND	ND	69.69 ± 2.33 ^a^	34.19 ± 0.83 ^b^	31.62 ± 0.44 ^c^
	3	ND	ND	ND	139.10 ± 3.00 ^a^	101.90 ± 3.51 ^b^	94.68 ± 3.05 ^c^
	5	ND	ND	ND	1.96 ± 0.25 ^a^	0.18 ± 0.04 ^b^	ND
	6	ND	ND	ND	ND	ND	1.73 ± 0.11 ^a^
	8	1.28 ± 0.75 ^d^	ND	7.44 ± 5.58 ^c^	32.72 ± 1.33 ^a^	13.04 ± 0.61 ^b^	9.71 ± 0.30 ^bc^
	9	ND	ND	ND	10.19 ± 0.34 ^a^	4.76 ± 0.40 ^b^	2.46 ± 0.11 ^c^
	10	26.28 ± 3.96 ^c^	29.59 ± 1.75 ^c^	57.62 ± 41.45 ^bc^	101.21 ± 6.13 ^a^	69.83 ± 2.35 ^b^	56.32 ± 0.93 ^bc^
	12	23.94 ± 21.17 ^b^	ND	ND	44.19 ± 0.85 ^a^	18.85 ± 2.01 ^b^	14.15 ± 0.25 ^bc^
	15	ND	ND	ND	6.11 ± 0.32 ^a^	1.90 ± 0.30 ^b^	1.55 ± 0.16 ^c^
	17	12.80 ± 1.75 ^c^	30.92 ± 5.84 ^b^	ND	36.93 ± 0.47 ^a^	11.96 ± 1.54 ^c^	9.75 ± 0.35 ^c^
	19	ND	ND	3.24 ± 0.36 ^d^	25.33 ± 0.60 ^a^	19.94 ± 1.03 ^b^	10.53 ± 0.10 ^c^
	20	65.01 ± 2.95 ^a^	68.65 ± 8.04 ^a^	37.89 ± 3.59 ^b^	21.15 ± 0.60 ^c^	24.23 ± 0.14 ^c^	13.86 ± 0.69 ^d^
	22	ND	46.52 ± 36.97 ^a^	12.37 ± 0.56 ^b^	24.41 ± 1.58 ^ab^	6.90 ± 1.46 ^b^	9.51 ± 0.35 ^b^
	23	82.66 ± 3.88 ^a^	71.87 ± 46.73 ^a^	51.96 ± 4.47 ^ab^	23.41 ± 2.13 ^bc^	11.68 ± 2.03 ^c^	16.72 ± 1.13 ^bc^
	24	22.39 ± 1.78 ^bc^	26.05 ± 17.72 ^bc^	17.17 ± 1.20 ^c^	48.10 ± 0.91 ^a^	50.18 ± 2.48 ^a^	32.92 ± 1.32 ^b^
	25	24.74 ± 2.06 ^d^	27.89 ± 4.08 ^d^	8.97 ± 0.53 ^e^	85.93 ± 2.02 ^b^	93.19 ± 7.50 ^a^	50.59 ± 1.14 ^c^
	26	62.75 ± 3.21 ^c^	94.67 ± 4.48 ^a^	61.79 ± 5.51 ^c^	82.98 ± 2.59 ^b^	57.48 ± 4.15 ^cd^	52.40 ± 1.63 ^d^
	27	78.64 ± 1.80 ^cd^	105.65 ± 4.28 ^b^	67.48 ± 3.28 ^e^	144.40 ± 4.29 ^a^	73.97 ± 6.88 ^de^	81.54 ± 1.06 ^c^
	33	81.17 ± 4.37 ^b^	134.72 ± 6.54 ^a^	62.52 ± 2.61 ^c^	37.38 ± 3.36 ^d^	21.74 ± 1.13 ^e^	25.60 ± 1.08 ^e^
	34	49.89 ± 3.05 ^c^	93.62 ± 4.73 ^a^	41.38 ± 2.69 ^d^	77.16 ± 2.21 ^b^	38.37 ± 3.73 ^d^	40.05 ± 0.85 ^d^
	35	65.39 ± 3.99 ^cd^	90.09 ± 6.35 ^b^	55.17 ± 5.51 ^d^	123.33 ± 4.00 ^a^	54.84 ± 13.52 ^d^	68.87 ± 1.11 ^c^
**Subtotal**		**596.91 ± 40.90 ^d^**	**820.24 ± 35.85 ^b^**	**485.01 ± 23.31 ^e^**	**1144.37 ± 18.86 ^a^**	**711.95 ± 28.26 ^c^**	**627.71 ± 8.24 ^d^**
**Total**		**993.71 ± 57.27 ^cd^**	**1364.05 ± 73.40 ^b^**	**881.16 ± 5.15 ^e^**	**1674.60 ± 25.45 ^a^**	**1058.83 ± 45.69 ^c^**	**945.17 ± 11.08 ^de^**

^(1)^ 1, platycoside G2; 2, platycoside G1(deapi-platycoside E); 3, platycoside E; 4, platycoside D; 5, 3”-*O*-acetyl platycoside E; 6, platycoside P; 7, platycoside I(deapi-polygalacin E); 8, deapi-platycodin D3; 9, 2”-*O*-acetyl platycoside E; 10, platycodin D3; 11, platyconic acid A3; 12, 3”-O-acetyl platycodin D3; 13, 3”-*O*-acetyl platyconic acid A3; 14, polygalacin D3; 15, deapi-2”-*O*-acetyl platycodin D3; 16, 3”-*O*-acetyl polygalacin D3; 17, 2”-*O*-acetyl platycodin D3; 18, 2”-*O*-acetyl platyconic acid A3; 19, deapi-platycodin D2(platycoside A); 20, platycurodin D; 21, 2”-*O*-acetyl polygalacin D3; 22, deapi-3”-*O*-acetyl platycodin D2; 23, 3”-*O*-acetyl platycurodin D(platycodin L); 24, platycodin D2; 25, platycodin D; 26, 3”-*O*-acetyl platycodin D2; 27, 3”-*O*-acetyl platycodin D(platycodin C); 28, 3”-*O*-acetyl polygalacin D2; 29, 2”-*O*-acetyl platyconic acid C(platyconic acid D); 30, 3”-*O*-acetyl polygalacin D; 31, platyconic acid A; 32, 3”-*O*-acetyl platyconic acid A; 33, 2”-*O*-acetyl platycurodin D(platycodin K); 34, 2”-*O*-acetyl platycodin D2(platycodin V); 35, 2”-*O*-acetyl platycodin D(platycodin A); 36, 2”-*O*-acetyl polygalacin D2; 37, 2”-*O*-acetyl polygalacin D; 38, 2”-*O*-acetyl platyconic acid A.

**Table 2 molecules-27-00107-t002:** MS parameters for saponin analysis.

Items	Conditions
Ion source gas (psi)	50
Curtain gas (psi)	30
Ion source temperature (°C)	450
Declustering potential (V)	80
Collision energy (V)	15 ± 10
Spray voltage (V)	5500
CE spread (V)	10
Ionization mode	Positive
Mass range (*m/z*)	100–2000

**Table 3 molecules-27-00107-t003:** UPLC separation conditions for saponin analysis.

Items	Conditions
Column	CORTECS ^®^ UPLC ^®^ T3, 2.1 × 15 0 mm, 1.6 μm
Pre-column	CORTECS ^®^ UPLC ^®^ Vanguard T3, 2.1 × 50 mm, 1.6 μm
Column temperature	30 °C
Detector	190–400 nm (representative wavelength, 203 nm)
Mobile phase	A: 0.1% Formic acid in water B: 0.1% Formic acid in acetonitrile
Injection volume	1 µL
Flow	0.35 mL/min
Running time	60 min
Gradient condition	0–10 min (15–25% B), 10–40 min (25–50% B), 40–45 min (50–15% B), 50–60 min (15% B)

## Data Availability

The data that support the results and findings of this study is available from the corresponding author upon request.

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
