# Peer review of "Characterization of Saponins from Various Parts of Platycodon grandiflorum Using UPLC-QToF/MS"

_molecules, 2021, doi:10.3390/molecules27010107_

Round 1
Reviewer 1 Report
There is a few explanation why the aim of the study are saponins. What is the real interest to perform a study on this class of compounds. It should better explained. The english language and style should be revised and improved, there are many sentences that do not make sense and/or are hard to understand.
Some comments:
Title: please chage "used parts"
Line 12 - The scientific name should be given in italic.
Line 16 - morphological parts
Line 37 - Reformualte sentence
Line 39 - Plants synthesized different bioactive....
Line 49 - 61 - Should be in the discussion part
Line 64 - Does not make sense
Line 107 - Improve the quality of Figure 1
Line 110 - Figures 2 and 3 show the presence of other compounds (peaks) besides saponins. Why these compounds were not identified?
Line 144 - The peels should have been analysed to better understand the losses verified in this study.
Line 145 - THe authors mention that the aerial aprts contain phenolic compouds but these are not identified or quantified.
Line 166 - Please give the exact %. Be concise. The same is applied throught the discussion of results.
Line 292 - month of collection? The different morphological parts should be collected from the same area and day. It does not make sense to compare samples collected in different areas and years. There are too many variables that contribute to the phytochemical composition of samples (soil, climate, watering, etc).
Line 299 - WHat is the ratio sample/solvent of extraction?
Table 2 - It is hard to read the detector conditions.
Line 329 - A better explanation should be given about the calibration curve using the internal standard. Also, if the authors had the aglycones of saponins why they did not perform the quantification using the respective standards?
Line 353 - Scientific name in italic
Author Response
Please see the corrections in the attached manuscript.

Reviewer 2 Report
About the quantitative analytical method for saponins, Line 327, how did the author use 9 compounds to detect other 39 saponins?
At detection wavelength 203nm, can HPLC show any peaks?
Table 3, how about the MS/MS parameters for MSn analysis?
The resolution of Figure 1-3 is too low, I hardly see what the real words in the figures.
How about the Methodology of analytical method?
How about the mass deduction process, with any regulation of mass fragmentation? in Figure?
Round 2
Reviewer 1 Report
The manuscript quality has improved and, in my opinion, is suitable for publication in its current form.